# A specific innate immune response silences the virulence of *Pseudomonas aeruginosa* in a latent infection model in the *Drosophila melanogaster* host

Jing Chen[1], Guiying Lin[1,2,3], Kaiyu Ma[1], Zi Li[1], Samuel Liégeois[1,2,3], Dominique Ferrandon [1,2,3]*

1 Sino-French Hoffmann Institute, School of Basic Medical Sciences, Guangzhou Medical University, Guangzhou, China, 2 Université de Strasbourg, Strasbourg, France, 3 Modèles Insectes de l'Immunité Innée, UPR 9022 du CNRS, Strasbourg, France

* D.Ferrandon@unistra.fr

**Data Availability Statement:** The primary data of this work have been deposited on Figshare (doi: 10.6084/m9.figshare.2591200).

## Abstract

Microbial pathogenicity often depends on the route of infection. For instance, *P. aeruginosa* or *S. marcescens* cause acute systemic infections when low numbers of bacteria are injected into *D. melanogaster* flies whereas flies succumb much slower to the continuous ingestion of these pathogens, even though both manage to escape from the gut compartment and reach the hemocoel. Here, we have developed a latent *P. aeruginosa* infection model by feeding flies on the bacteria for a short period. The bacteria stably colonize internal tissues yet hardly cause any damage since latently-infected flies live almost as long as non-infected control flies. The apparently dormant bacteria display particular characteristics in terms of bacterial colony morphology, composition of the outer cell wall, and motility. The virulence of these bacteria can however be reactivated upon wounding the host. We show that melanization but not the cellular or the systemic humoral response is the predominant host defense that establishes latency and may coerce the bacteria to a dormant state. In addition, the lasting activation of the melanization responses in latently-infected flies provides a degree of protection to the host against a secondary fungal infection. Latent infection by an ingested pathogen protects against a variety of homologous or heterologous systemic secondary infectious challenges, a situation previously described for the endosymbiotic Wolbachia bacteria, a guard against viral infections.

## Author summary

Environmentally ubiquitous bacteria have acquired extensive abilities to adapt to variable environments, bestowing to some of them the potential to become opportunistic pathogens. This may translate into distinct infection modes according to the route of entry. Whereas *Pseudomonas aeruginosa* is considered to have two major modes of infection, acute by planktonic cells or chronic through the establishment of biofilms, we report here

**Funding:** This work has been supported by Guangzhou Medical University, Université de Strasbourg, and CNRS, and also the China High-end Foreign Talent program (DF), the 111 Plan (Overseas Expertise Introduction Project for Discipline Innovation; D18010 to the Sino-French-Hoffmann Institute), the Incubation Project for Innovative Teams of Guangzhou Medical University (ZL, DF), the Open Project from State Key Laboratory of Respiratory Diseases (ZL, DF), the China Postdoctoral Foundation Project (Postdoctoral Innovation Talent Support Program of China 2017M612634 to JC), and the Special Fund for Scientific and Technological Innovation Strategy of Guangdong Province (2018A030310180 to JC). The funders had no role in study design, data collection and analysis, decision to publish, or preparation of the manuscript.

**Competing interests:** The authors have declared that no competing interests exist.

a novel type of infection whereby ingested bacteria escape from the digestive tract and silently colonize tissues as single cells without strongly affecting the lifespan of the *Drosophila* host. The bacteria appear to be dormant, a feature shared with persister cells that elude the action of antibiotics. They are characterized by distinct bacterial and colony morphologies, cell surface and motility properties. Their virulence program can nevertheless be reactivated spontaneously or upon injury. We also report that an important host defense of invertebrates, melanization, is activated upon escape of the bacteria into the internal milieu. This activation not only promotes the dormancy of the colonizing bacteria but also protects the host to some degree against secondary infections. As *P. aeruginosa* is a member of the microbiota of a sizable fraction of human populations, these discoveries may become medically relevant.

## Introduction

*Pseudomonas aeruginosa* is a Gram-negative bacterium that can cause acute or chronic infections in immune-compromised individuals and patients suffering from AIDS, burn-wounds, cystic fibrosis or chronic obstructive pulmonary disease [1]. Most acute *P. aeruginosa* infections can be controlled by antibiotic drugs and immune responses. However, some pathogens can remain for long periods within the host due to a defective immune response, effective immune evasion strategies or adaptation to antibiotic treatments of *P. aeruginosa* [2]. In addition to genetically encoded resistance, a fraction of bacteria in antibiotic-sensitive strains may tolerate the antibiotic treatments and are referred to as persister cells. These resilient bacteria can lead to the relapse of the bacterial infection when the host is weakened or when the antibiotic treatment is discontinued [3]. Many studies have attempted to identify the mechanisms of persister cell formation focusing on bacterial cells *per se*, such as autotoxins liberated from toxin-antitoxin systems resulting in growth arrest, metabolic regulation and protein homeostasis implicated in diauxic shift, or reduced production of ATP [2]. Whereas many leads are being investigated, the definitive *in vivo* mechanism remains rather poorly understood [2]. It is however also important to know how host immunity works to recognize and control bacteria, not only by killing them but also by influencing their physiology and implementation of virulence programs.

  *Drosophila melanogaster* is a powerful animal model to study the innate immune response since it lacks a mammalian-like adaptive immune system. It has been used to study the host response to *P. aeruginosa* for more than 50 years [4, 5]. In flies, immunity against systemic bacterial or fungal infections relies on three arms: i) the systemic humoral immune response, regulated by the NF-κB-type Immune deficiency (IMD) and Toll pathways producing antimicrobial peptides (AMPs) [6], phagocytosis mediated by mammalian macrophages-like plasmatocytes [7] and melanization that results from the activation of dedicated serine proteases cascades [8]. Melanization involves a series of enzymatic reactions that are catalyzed by phenol oxidases, found as pro-proteins in the hemolymph, prophenol oxidases (PPOs). PPOs are cleaved into active POs by proteases such as Sp7 or Hayan, the latter being supposedly required for the activation of the three *Drosophila* PPOs [8–11]. In larvae, PPOs are stored in crystal cells and are released in the hemolymph by a JNK-dependent rupture of their cytoplasmic membranes [12]. In adults, the situation is less clear. Whereas PPO2 is expressed in some 8% of adult hemocytes [13], whether PPO2 forms crystals as observed in larvae remains to be firmly established. In addition, as noted above, PPO1 and PPO2 are also found circulating in the hemolymph. They can mediate melanization at the site of wounding upon

proteolytic cleavage of their Pro-domains and are also required for a microbial killing activity that remains poorly understood [8] and that may be mediated by Reactive Oxygen Species (ROS) [14, 15].

*P. aeruginosa* is strongly virulent in acute systemic infection models and a few bacteria injected into the fly body cavity and its open circulatory system kills host flies within a few days despite the induction of a humoral systemic immune response mediated by the IMD pathway and by a second NF-κB pathway, the Toll pathway [16]. Of note, *P. aeruginosa* in this acute infection model appears to elude activating the melanization response in a process that requires the *Drosophila* complement factor Tep4 [17]. This pathogen has also been reported to inhibit to some extent the IMD pathway [18].

Flies in their natural environment feed on rotten fruits, a microbe-rich environment in which *P. aeruginosa* usually survives in well [19]. A continuous feeding infection model has been established in *Drosophila* [20]. *P. aeruginosa* were retrieved very early on from the hemolymph, implying that some bacteria can cross rapidly the gut barrier. Interestingly, *P. aeruginosa* displayed impaired pathogenicity as compared to a systemic infection and flies succumbed in about a week [20]. More importantly, the bacterial titer in the hemolymph remained at a steady low level until mid-infection and then started increasing exponentially, leading to the rapid demise of the flies that succumbed to bacteremia [17]. Several immune responses contribute to control the pathogenicity of ingested *P. aeruginosa*, such as IMD pathway activation in the gut, hemocyte-mediated phagocytosis, and the systemic immune response mediated to some extent by AMPs released by the fat body a composite analogue of the vertebrate liver and adipose tissue. The cellular immune response limits proliferation of bacteria that have crossed the gut barrier by elimination through phagocytosis [20]. Based on this, we do not definitely know whether the steady bacterial load in the hemolymph at the beginning is due to a balance between bacteria entrance and killing, although this is highly likely. Moreover, a role for intestinal bacteria in participating to the ultimate killing of the host cannot be fully discounted, for instance if the gut barrier were disrupted at a late stage of infection.

Here, we have attempted to simplify the continuous ingestion infection model by allowing flies to feed on *P. aeruginosa* for only two days, a period during which the bacteria that have crossed the gut barrier and are found in the hemolymph appear to be relatively quiescent and easily controlled by the immune system. We report that even though these bacteria are rapidly cleared from the hemolymph within three days, other escaping *P. aeruginosa* bacteria manage to colonize the host tissues and remain dormant for the lifespan of the fly causing hardly any damage, unless sporadically reactivated. We study the different features of these dormant bacteria and establish that the *Drosophila* immune systems play an essential role in the establishment of quiescence in tissue-associated bacteria, with a major role played by melanization through cleaved PPOs, Sp7, and to a lesser extent Hayan.

## Results

### Establishment of a *P. aeruginosa* latent infection model in *Drosophila*

Whereas our usual intestinal infection model relies on continuous feeding on a pad containing PAO1 in a sucrose/BHB solution, we attempted to expose the flies for only a limited period to the bacterially-laced-solution at 18˚C. To ensure that no bacteria remained in the gut, this initial period of feeding on the bacterial solution was followed by a 4-day period during which flies fed on a gentamicin sucrose solution; this antibiotic poorly crosses the intestinal barrier [21,22]. Afterwards, the flies were kept also at 18˚C on a pad soaked with a sucrose solution (Fig 1A). Whereas flies feeding continuously on PAO1 usually succumb within 10 days at

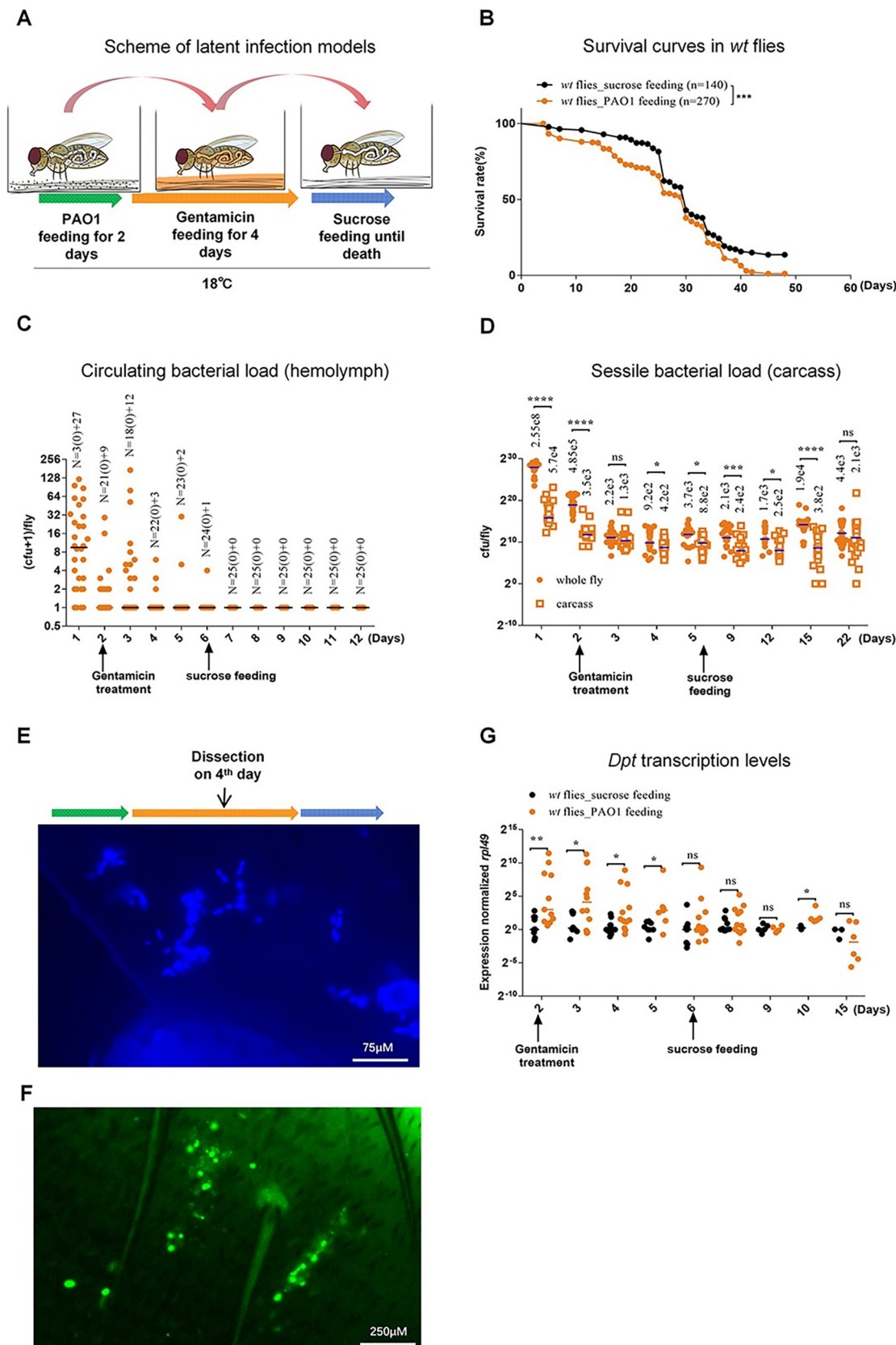

**Fig 1. *P. aeruginosa* that have escaped from the gut lumen into the body cavity lose pathogenicity and may become dormant in *Drosophila*.** (A) Scheme of *P. aeruginosa* latent infection model in *Drosophila*. Flies were fed with 10 $OD_{600}$ PAO1 suspended in 100mM sucrose solution with 10 percent Brain-Heart Infusion broth for 2 days and then PAO1 in gut lumen were killed by feeding sucrose solution containing 100μg/mL gentamicin for another 4 days, leaving alive only the bacteria that had crossed the digestive tract. Then flies were fed sucrose solution until death. Flies were put at 18°C throughout the infection process. (B) Survival curves after latent infection. (C) Bacterial titer from collected hemolymph. Each dot indicates the bacterial number collected from a single fly. N, the number of flies, is indicated above the data for each time point; the first number corresponds to the number of flies for which no bacteria were retrieved (0) and the second number corresponds to the number of flies for which bacteria were retrieved. (D) Time course of bacterial titer in whole flies (filled circles) and carcass (empty squares). Each dot indicates the bacterial number retrieved from a single fly or a single carcass. The numbers above each column indicate the median value of the bacterial titer in each column. (E-F) *P. aeruginosa* visualization in fly tissue. (E) *P. aeruginosa* PAO1 was visualized by HOECHST staining. (F) *P. aeruginosa* PA14 was stained with an antibody raised against it. (G) Activation of the IMD pathway upon ingested *P. aeruginosa* infection as monitored by RTqPCR of *Diptericin* transcripts normalized to *rpl49* transcript levels of each sample. (B-G). All experiments here were performed three times independently, and data were pooled together (B, C, D, G). The bar indicates the median in each column (C, D and G). Statistical analysis was done by Logrank (Mantel-Cox test) in (B) or by Mann-Whitney test in (D and G).

18°C (S1A Fig), flies on this new regimen of short period of ingestion of PAO1 survived much longer, with a LT50 (time taken by 50% of the flies to succumb to infection) of about 25 days. They actually survived this treatment almost as long as sucrose-only fed controls, albeit they nevertheless succumbed significantly earlier (Fig 1B). We then checked the microbial burden in the hemolymph of flies exposed for two days to PAO1 since *P. aeruginosa* is known to cross the gut barrier within hours [17,20]. Interestingly, we detected tens of bacteria during the first three days of the protocol before the hemolymph titer progressively decreased to zero (Fig 1C). Next, we checked whether the bacteria might have colonized the tissues by measuring the titer in whole flies or in carcasses, that corresponds to flies in which most of the digestive tract and the ovaries had been surgically removed. Whereas the difference between whole flies and carcass differed by more than 10-fold in the first two days of infection when flies were feeding on PAO1 reflecting the presence of bacteria in the gut lumen, the difference was much less pronounced later on (Fig 1D). We checked using GFP-labeled PAO1 that the bacteria were essentially removed from the gut lumen by the gentamicin treatment (S1B and S1C Fig), also in keeping with the previous observation that *P. aeruginosa* bacteria are cleared from the gut when subsequently fed on sucrose for 24 hours [20]. Thus, these data establish that a few thousand bacteria do colonize the fly tissues after having crossed the gut barrier during the first two days of the infection protocol. The bacterial burden appeared to remain stable in most flies, at least until the last day tested, day 22. Of note, as for other tissues and organs that remain in the carcass, bacteria can also associate with the crop and gut compartments, but essentially from the outside of the gut, *e.g.*, in visceral muscles, since bacteria within the digestive tract have been cleared by the gentamicin treatment and to a sizable extent by continuous feeding on sucrose solution alone instead of gentamicin solution (S1B Fig) [23]. We were actually able to detect bacteria under the fly cuticle either by Hoechst dye staining (Fig 1E) or using antibody staining (Fig 1F). We also monitored the induction of the systemic immune response by monitoring the steady-state levels of *Diptericin* mRNAs by RTqPCR. While we detected a significant induction of *Diptericin* in the first few days of the infection that eventually subsided (Fig 1G), the level of induction of IMD-regulated AMP genes was much lower than that observed during a systemic infection (S1D Fig). By monitoring transgenic flies that express a GFP protein under the control of the *Diptericin* promoter, we noted that a few flies occasionally exhibited a signal as strong as that induced during a systemic immune response induced in a septic injury model (S1E Fig). Strikingly, the *pDipt-GFP*-positive flies were fated to die shortly after the detection, suggesting that in those flies a full-blown bacteremia was underway. In contrast, the *pDipt-GFP*-negative flies survived suggesting that PAO1 does not proliferate in those flies. Thus, the virulence of PAO1 can be spontaneously reactivated in a few flies. We conclude that

under conditions of limited ingestion, PAO1 colonizes fly internal tissues without causing a systemic infection in most cases and therefore refer to this model of infection as a latent infection model: the bacteria present in the hemocoel appear to be dormant.

We had previously reported that at 25˚C flies fed on bacteria for up to three days and then subsequently fed on sucrose solution alone (no gentamicin treatment) did not succumb to infection within two weeks as when feeding continuously on the bacterial solution [20]. In retrospect, these experiments also suggest that gentamicin treatment may not be essential to establish latency and that a decreased supply of bacteria in the gut lumen is the important parameter. Indeed, the titer of bacteria in the hemolymph decreased already after 24 hours on sucrose solution (day 3, S1F Fig) whereas PAO1 was still colonizing the carcass as efficiently as after gentamicin treatment (S1G Fig).

### PAO1 bacteria display distinct properties depending on the infection route

Given the strikingly different properties of virulence displayed by PAO1 depending on the infection route (injection [death within a couple of days] *vs.* ingestion) and the exact protocol (death in some 7 days upon continuous feeding on PAO1 *vs.* some 40 days in the latent infection model), we checked whether differences in bacterial morphology could be detected. To this end, we first put flies to feed on RFP-expressing PAO1 bacteria and two days thereafter injected GFP-expressing PAO1 in the body cavity (hemocoel). Under the cuticle, the injected green bacteria appeared elongated and much slenderer than the ingested red bacteria that appeared plumper (S2A Fig). These observations were confirmed by electron microscopy on bacteria retrieved in the hemolymph of flies that had either ingested PAO1 (Fig 2A) or been injected with PAO1 (Fig 2B). Strikingly, bacteria grown in BHB present an ellipsoid morphology and are much shorter but with a larger diameter (Fig 2C). Of note, the bacteria in the gut lumen appeared under light microscopy not to be elongated and had a shape more reminiscent of that of BHB-grown bacteria than that of bacteria observed in the hemolymph (S2B Fig).

We also assessed whether PAO1 bacteria displayed the O5 LPS antigen by staining bacteria *in situ* using a specific antibody [24,25]. Injected PAO1 bacteria found in the tissues or retrieved from the hemolymph were expressing the O5 antigen (Fig 2D left panels). In contrast, ingested RFP-expressing PAO1 bacteria adhering to tissues did not express the O5 antigen (Fig 2D bottom right panel). Interestingly, the few bacteria that are found in the hemolymph at the onset of the intestinal infection did express the LPS antigen (Fig 2D top right panel).

When plated, the colonies harbored also distinct shapes. The bacterial colonies derived from injected bacteria appeared to be larger with indistinct boundaries and a tendency toward clustering with other colonies. In contrast, bacteria retrieved from the carcass of latently-infected flies appeared smaller with well-delimited contours. PAO1 grown in BHB liquid medium yielded colonies of intermediate sizes with fuzzy shapes (Fig 2E). We next tested the motility of PAO1 bacteria in a swimming assay: whereas bacteria that had been injected and retrieved from the *Drosophila* host were motile, the ones extracted from latently-infected flies remained immobile (Fig 2F).

Dormant bacteria have been reported to be more tolerant to antibiotics treatment than metabolically active bacteria. We therefore first treated flies that had been injected a day before with PAO1 by injecting either PBS, tobramycin or levofloxacin at a concentration optimized in the injection model to provide a large degree of protection against the infection (S2C–S2C' Fig). The injected bacteria were clearly sensitive to tobramycin and somewhat sensitive to levofloxacin (Fig 2G'); since levofloxacin at the concentration used protects flies from injected PAO1 as well as tobramycin (S2C–S2C' Fig), it follows that the former antibiotics also affects

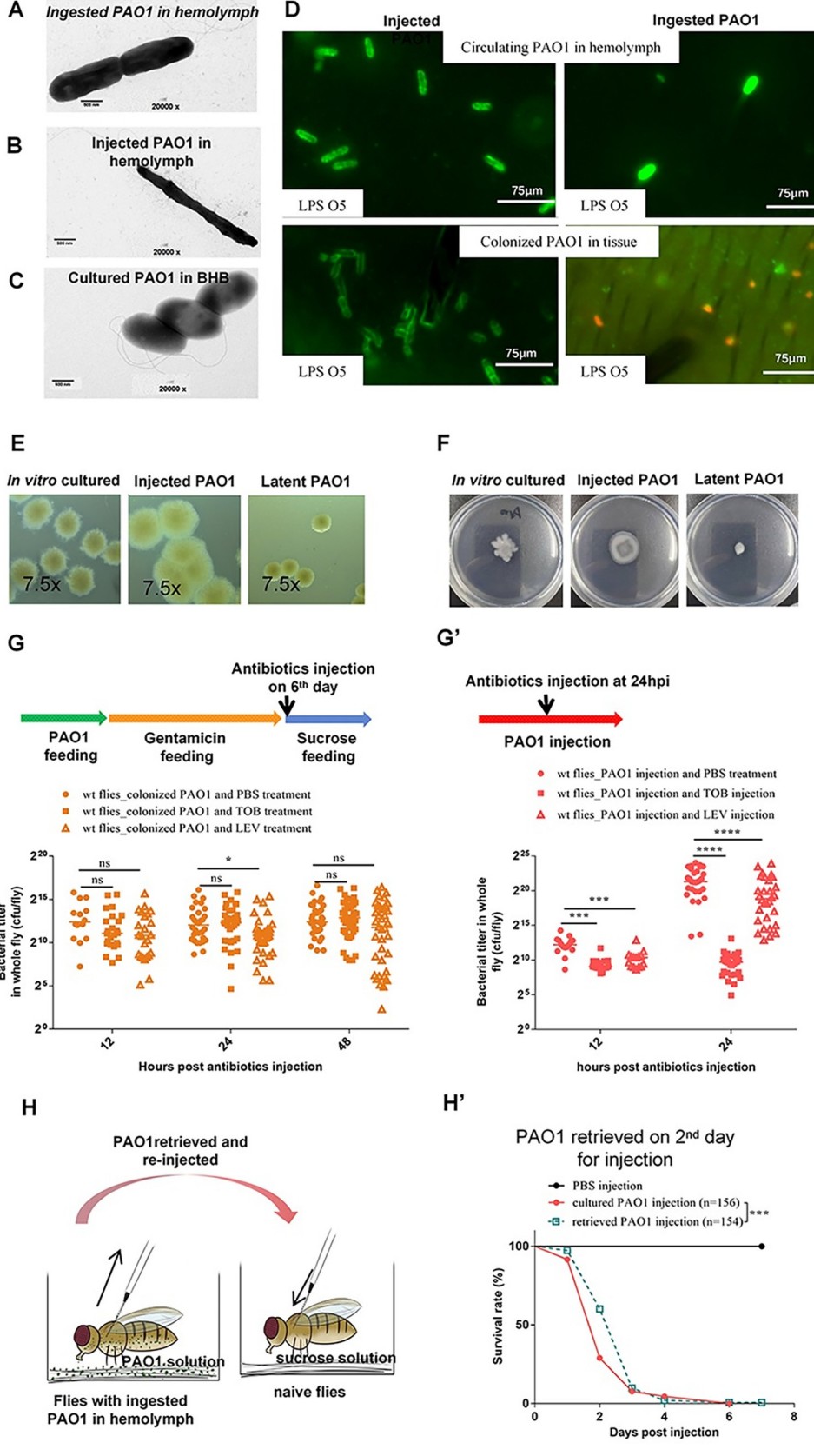

**Fig 2. *P. aeruginosa* bacteria colonizing tissues in the body cavity exhibit distinct traits from injected ones. (A-C)** Morphology of *P. aeruginosa* in the latent infection model (A), in the injection model (B), and cultured in BHB (C). Hemolymph was collected from flies with injected or ingested PAO1 respectively and the bacteria were observed by transmission electron microscopy. Scale bars are 500 nm. **(D)** LPS O5 antigen staining (green) of RFP-expressing bacteria retrieved from hemolymph (top panels) or from tissues (bottom panels). Note the absence of staining in bacteria colonizing the tissues in the latent infection model. **(E)** Colonies of bacteria retrieved from infected flies (at Day6 for latently-infected flies and at Day2 for injected flies), as indicated, forming on Congo RED agar plates. *In vitro* cultures of PAO1 were in BHB medium. **(F)** Motility assay of bacteria retrieved from infected flies, as indicated, on low percentage agar plates. **(G and G')** Sensitivity of bacteria *in vivo* to 4.6 nL-injected tobramycin (TOB: 16mg/mL) or levofloxacin (8mg/mL) measured by assessing the bacterial load of single whole flies at the indicated times after antibiotics injection in the latent (G) or injection (G') infection model. **(H)** Scheme of pathogenicity experiment for PAO1 retrieved from latently-infected flies. Hemolymph was collected from flies with ingested PAO1 at Day 2 post feeding and injected into naive flies in couples. **(H')** Survival curves of naïve flies to injected cultured *P. aeruginosa* (red curve) or *P. aeruginosa* retrieved from latently-infected flies (orange curve). All experiments were performed three times independently; data were pooled (G, G', and H'). The bar indicates the median in each column (G and G'). Statistical analysis was done using the Kruskal-Wallis test with Dunn's post-hoc test (G and G'), or by Logrank (Mantel-Cox test) in (H').

the virulence of the injected bacteria in the septic injury model in addition to its moderate effect on the bacterial titer. In contrast, when flies in which the latent infection had been implemented six days earlier were injected with either antibiotic, the bacteria that had colonized the fly tissues appeared to tolerate well both antibiotics (Fig 2G) since the bacterial load was not altered. We have also checked that the ingestion of levofloxacin, as opposed to its injection, protected flies from injected PAO1 (S2D Fig). Nevertheless, PAO1 bacteria in latently-infected flies that had been fed antibiotics on day6, after gentamicin treatment, were tolerant to these antibiotic treatments, in contrast to those injected as witnessed by the decreased bacterial burden observed in PAO1-injected flies (S2D'–S2D" Fig).

One possibility to account for these diverse observations would be that bacteria are selected for irreversibly attenuated virulence during the establishment of the latent infection, for instance, through mutation. We therefore collected bacteria from latently-infected flies at different time points, either from hemolymph (very few bacteria) or retrieved from the crushed carcass and then injected them directly into naive recipient flies. The flies injected with bacteria passaged in the fly using the latent infection protocol at days 2, 7, or 20 killed the naive flies rapidly, at a pace that was similar as when flies were directly injected with bacteria grown in BHB liquid culture, with most flies being dead three days after infection (Figs 2H–2H' and S2E–S2E'). We conclude that bacteria in the latent infection model are not irreversibly committed to impaired virulence and do not undergo selection at the genomic level.

## The host innate immunity is required to prevent the pathogenicity of PAO1 in the latent infection model

The outcome of an infection results from the interactions between the pathogen and its host. We first monitored the survival rates of fly strains deficient for the diverse arms of the *Drosophila* immune response that had ingested PAO1 under the latent infection protocol. Flies deficient either for the cellular immune response due to an absence of the Eater phagocytic receptor or flies in which the IMD pathway cannot be activated succumbed faster than wild-type flies, with a LT50 of some 14 days (*vs.* about 21 days for wild-type flies) (Fig 3A). This increased virulence of PAO1 in immuno-deficient flies was mirrored in the bacterial burden measured on collected hemolymph from the second day onward and became highly significant on the 10[th] day (Fig 3B). It is likely that flies harboring a high PAO1 hemolymph titer are fated to die rapidly in the following day(s). We also ablated hemocyte phagocytic function by saturating the phagocytic machinery by the injection of latex beads at day 10 of the protocol and monitored the survival of those flies until death. As compared to PBS-injected controls,

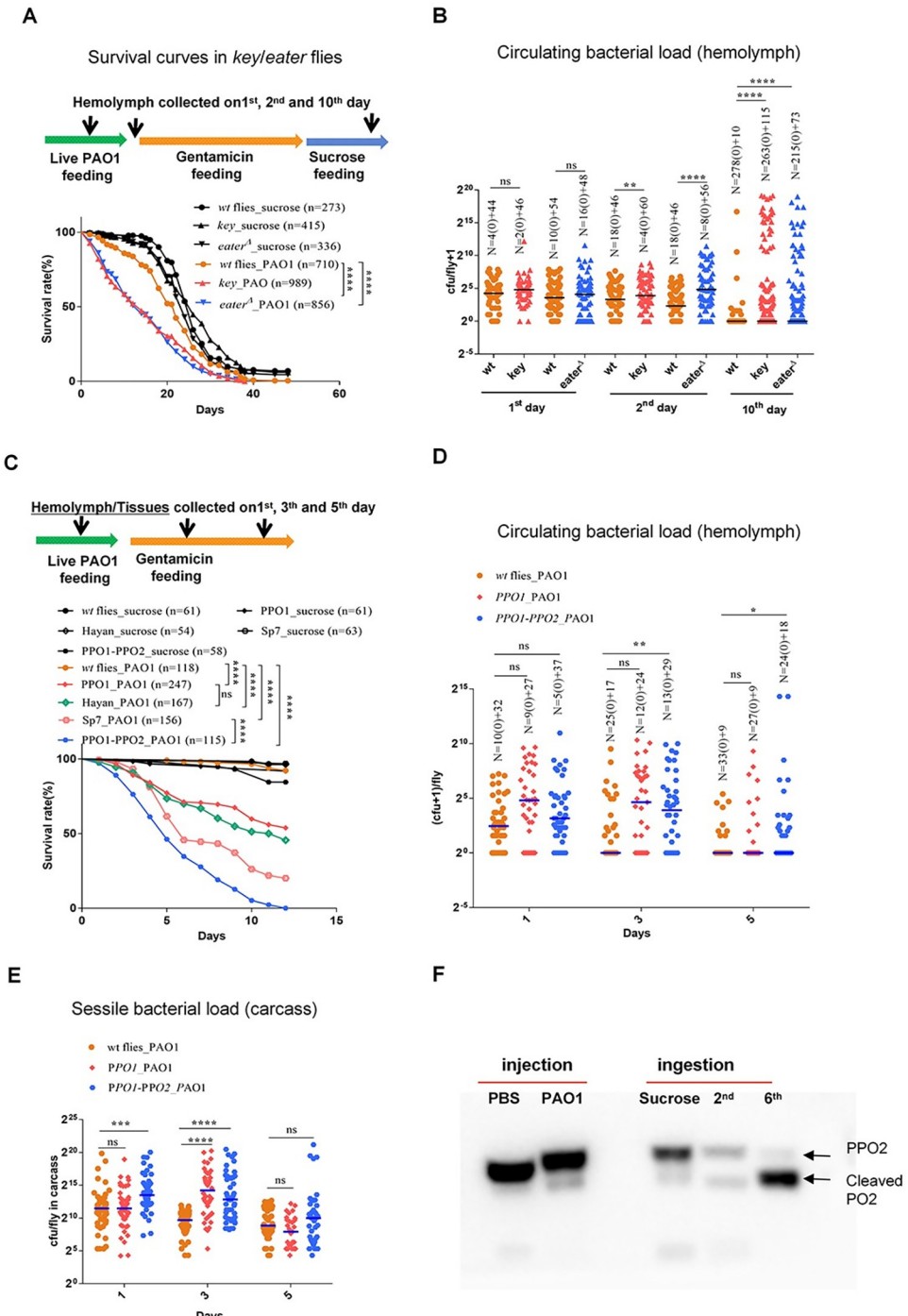

**Fig 3. Melanization is the preponderant host defense that initiates dormancy in the bacteria that have escaped from the digestive tract. (A)** Survival curves for *key* and *eater* deficient flies. **(B)** Bacterial titer in the hemolymph collected from *key* and *eater* deficient flies. Each dot corresponds to the bacterial load of a single fly. **(C)** Survival curves of melanization-deficient flies: *Hayan*, *ΔPPO1-ΔPPO2*, and *Sp7*. **(D)** Bacterial titer in the hemolymph collected from wild-type flies or mutant for *phenol oxidase* genes. **(E)** Bacterial titer of single fly carcasses of prophenoloxidase-deficient flies. **(F)** PPO cleavage in different infection models as revealed by Western blot analysis using a pan-PO reactive antibody. All experiments here were performed three times independently and data were pooled together (A-E). The bar indicates the median in each column (B, D and E). Statistical analysis was done using Logrank (Mantel-Cox test) in (A and C), and Mann-Whitney test in (B), Kruskal-Wallis with Dunn's post-hoc test in (D and E).

latently-infected flies injected with latex beads displayed a shortened lifespan, but not an acute death profile in the few days that follow the injection, as would have been expected in the case of bacteremia (S3A Fig). Indeed, when we monitored the bacterial burden, we observed bacteria in the hemolymph of some flies and no significant increase in the bacterial load in the carcass (S3B Fig). Taken together, these data suggest that phagocytosis is required to limit the proliferation of the bacteria in the hemolymph when some tissue-colonizing bacteria become planktonic. Similarly, the IMD pathway might become activated when bacteria start dividing actively and release peptidoglycan fragments that are sensed by PGRP-LC and /or PGRP-LE.

As noted in the introduction, melanization is not efficiently triggered in the septic injury model and indeed *ΔPPO1-ΔPPO2* mutants succumbed at the same rate as wild-type flies (S3C Fig). We next tested several mutant lines affecting melanization, which all displayed a greatly increased sensitivity to ingested PAO1 in the latent infection model, with the exception of the *ΔPPO1*, *ΔPPO2* or *Hayan* single mutant strains for which 50% or less of the flies succumbed to the infection within two weeks (Figs 3C and S3D). The *ΔPPO1-ΔPPO2* and *Sp7* mutant lines exhibited the highest susceptibility with a LT50 value of respectively about four and a half and five and a half days (Fig 3C). Unexpectedly, *Hayan* flies were somewhat more resistant than these mutant lines, with a LT50 of about 10 days in this series of experiments. Of note, whereas *ΔPPO1-ΔPPO2*, and *Sp7* mutant flies succumbed faster than *eater* or *key* mutant flies, they nevertheless did not die as fast as in the injection model (Fig 2H'). Interestingly, the bacterial load in the hemolymph of *PPO* mutants increased moderately, even at day five when half of the *ΔPPO1-ΔPPO2* flies have succumbed to the infection (Fig 3D). As regards the bacterial burden in the carcass, it was significantly higher already on the first day of infection for *ΔPPO1-ΔPPO2* (Fig 3E). On day three, *PPO1* and *ΔPPO1-ΔPPO2* mutants displayed a higher PAO1 load in the carcass that was however no longer observed at day five. Note that in this case the correlation between measured bacterial load (Fig 3E) and susceptibility to infection assessed in survival experiments was imperfect since *ΔPPO1* mutants do not succumb as fast as *ΔPPO1-ΔPPO2* double mutants in this two-day interval (Figs 3C and S3D). The systemic immune response becomes activated when the bacteria start proliferating in the hemocoel [17,20,23]. We observed that the expression level of *Diptericin* was increased in the *PPO* mutants from the third day onward (S3E Fig), in keeping with the increasing bacterial load measured in the carcass and hemolymph at day 3 (Fig 3D and 3E).

We then assessed whether the PAO1 bacteria were still dormant in *ΔPPO1-ΔPPO2* mutants. We found that in this background, bacteria were susceptible to injected tobramycin and levofloxacin (S4A and S4B Fig). Interestingly, bacteria retrieved from latently-infected flies at day 4 formed two types of colonies, small ones evocative of those formed by dormant bacteria and larger ones similar to those formed by in vitro-grown bacteria or those retrieved from flies acutely infected by injected PAO1 (S4C Fig). 72% +/-29% bacteria in the tissues (analysis of 15 fields) were positive for O5-antigen staining (S4D Fig) and mostly had kept a rounded morphology with sometimes a few bacteria having transitioned to a bacillus shape.

In conclusion, while melanization is required early on to block the virulence of ingested PAO1 that have crossed the intestinal barrier, the PAO1 bacteria do not nevertheless develop a full virulence program in its absence, as reflected in the shallow survival curves, a transition to virulence apparent in the bacterial sensitivity to antibiotics and expression of the O5-antigen in a large fraction of bacteria found in the inner cavity while maintaining a rounded shape, and more importantly in the limited bacterial growth in the body cavity.

We further directly tested whether melanization is indeed activated during the latent infection. Melanization results from the activation of a proteolytic cascade that ultimately cleaves prophenol oxidases into active phenol oxidases. We therefore tested by Western blot whether the cleaved form of PPOs was detectable in the hemolymph of flies submitted to varied

immune challenges. Whereas we detected a limited partial cleavage of PPOs after injection of PAO1 or a cleavage of about 50% of PPOs at day two of the ingestion of PAO1, PPOs were nearly fully cleaved at day six of the latent infection in this series of experiments (Figs 3F and S3F). We also monitored the expression levels of *Hayan* and *Sp7* transcripts by RTqPCR. Whereas both genes were induced one or two days after a PAO1 injection (S3G Fig), such an induction was not detected in the early days (one to three) of the latent infection model with expression levels remaining at a basal level (S3H Fig).

In conclusion, our data suggest that the low levels of IMD pathway activation (Figs 1G and S1D) prevent the proliferation of bacteria in the hemolymph at the initiation of the colonization whereas melanization inhibits the growth of PAO1 adhering to tissues (Fig 3E). We then set out to analyze how the induced immune defenses affect the outcome of the infection or of additional infections.

## A secondary acute PAO1 infection is mitigated in latently-infected flies

First, we monitored the effect of performing a wound of the cuticle of flies that had ingested PAO1. As shown in Fig 4A, the flies started to succumb when an injury was performed on the second day, at the end of the feeding period on PAO1. Similar results were obtained upon collection of hemolymph or the injection of PBS, processes that all involve perforating the cuticle with a glass microcapillary. Interestingly, this effect was much milder when the wound was made on day six of the protocol, when intestinal bacteria have been cleared by gentamicin treatment. Finally, hardly any short-term lethality was observed when flies had started to ingest PAO1 for ten days prior to the wound (S5A and S5B Fig). These experiments suggest that bacteria in the gut lumen or possibly in the process of crossing the gut barrier are sensitive to the systemic effects of a "clean" injury [26] and might reactivate their virulence programs. Alternatively, but not exclusively, host defenses triggered by the ingestion of PAO1 may become highly effective only after day six of the protocol, an observation that mirrors the progressive induction of melanization as revealed by the partial cleavage of PPO2 observed at day two in contrast to the full cleavage revealed at day six (Figs 3F and S3F).

We next performed a supernumerary injection of GFP-labeled PAO1 on flies that had ingested RFP-labeled PAO1 for two days already. As shown in Fig 4B, the dually infected flies succumbed to this challenge, but at a slower pace than naive flies fed on sucrose for two days that were first injected in parallel with PAO1. The analysis of the bacterial burden in the hemolymph and in the carcass revealed that PAO1 did not grow as fast in dually-infected flies as in naive flies (Fig 4C and 4D). Of note, both the GFP and the RFP-labeled bacteria proliferated at a similar rate in the hemolymph (S5C Fig). We conclude that the ingestion of PAO1 confers a partial protection against a systemic PAO1 infection. This protection was still observed in *eater* and *key* mutants (S5D Fig). Of note, the apparently higher resistance of *eater* mutants than *key* mutant flies fed on sucrose to a PAO1 injection cannot be directly compared as the genetic backgrounds of these flies differ.

A further control experiment was performed in which we compared the protection conferred by the ingestion of live vs. dead PAO1 (Fig 4E). This experiment revealed that the ingestion of live bacteria protected the host flies much more effectively than the ingestion of heat-killed bacteria, which are unlikely to escape from the digestive tract. The modest yet significant protection, a few hours, conferred by the ingestion of dead bacteria is likely mediated by the mild induction of the IMD pathway and not by the cellular response as revealed by the analysis of *key* (in which the protection conferred by dead bacteria ingestion was abolished) and *eater* mutants (S5E Fig). In keeping with these data, the ingestion of killed bacteria lead to a significant induction of the IMD pathway readout *Diptericin* 36–48 hours after ingestion (S5F Fig).

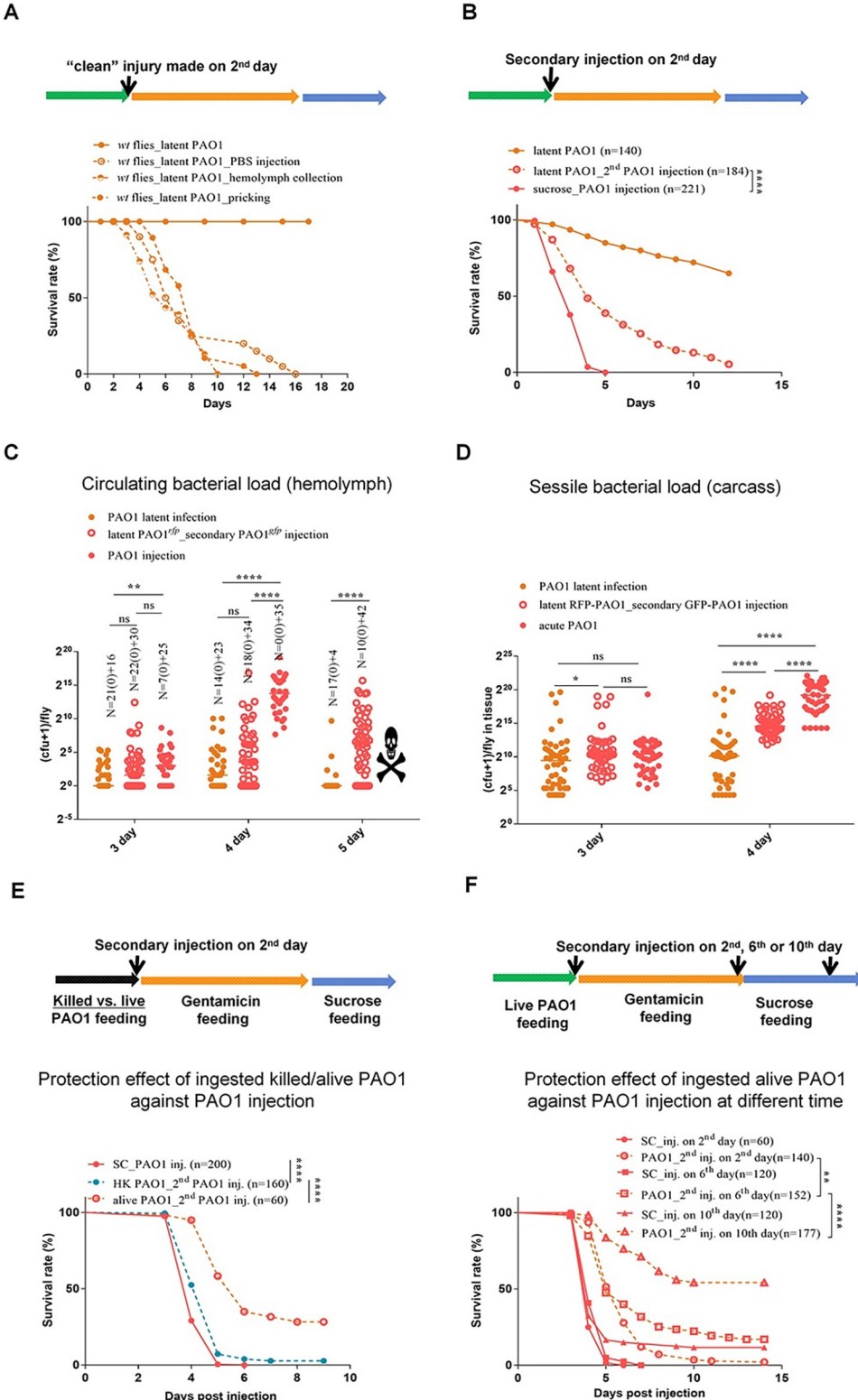

**Fig 4. The immune responses elicited by ingested *P. aeruginosa* provide protection against a secondary *P. aeruginosa* injection.** (**A**) Injury on flies at day 2 activates full virulence of ingested *P. aeruginosa*, as monitored in survival experiments. Injuries were performed either by pricking with a sharp needle, or by collecting hemolymph or injecting PBS with a micro-pipette. (**B**) Survival curves of flies submitted to a secondary infection by injecting cultured

RFP-labeled PAO1 to flies that had either ingested GFP-labeled PAO1 or a control sucrose solution. **(C-D)** Time course of bacterial burden of flies infected as in (B); the bacteria were collected either from the hemolymph or from carcasses of single flies. **(E)** The ingestion of heat-killed *P. aeruginosa* elicits a mild protection against injected PAO1 as compared to that induced by the ingestion of live *P. aeruginosa*, as monitored in survival experiments. **(F)** The protection afforded by the ingestion of live PAO1 becomes stronger as the secondary injection challenge is performed later and later. The protection progressively increases when the secondary injection is performed at day 6 and then day 10, as compared to the initial day 2 challenge when bacteria have not been cleared from the gut lumen. All experiments here were performed three times independently (4A excepted, one time) and data were pooled together (A-F). The bar indicates the median in each column (C and D). Statistical analysis was done using the Logrank test (Mantel-Cox test) in (A, B, E, and F) and using the Kruskal-Wallis test with Dunn's post-hoc test in (C and D).

Of note, UV-killed or chemically-fixed PAO1 provided a somewhat higher level of protection, by half a day, against injected PAO1 than heat-killed bacteria (S5G–S5G' Fig).

Interestingly, the protection of the host against a PAO1 systemic infection conferred by its prior ingestion was progressively much more pronounced when the supernumerary injection was performed at respectively days six and ten of the latent PAO1 infection protocol (Fig 4F). Thus, whereas the IMD-dependent host defense is triggered by bacteria present in the gut lumen, it appears that the protection conferred against systemic PAO1 infection by previously ingested bacteria correlates with the presence of bacteria inside the body cavity, that is bacteria colonizing tissues that do not appear to actively proliferate. The live bacteria may provide a partial protection against a secondary acute infection by stimulating another arm of the innate immune response. Alternatively, these bacteria may communicate with the incoming bacteria and modulate the expression of their virulence program(s). In the following, we address the first possibility by testing a variety of secondary infections.

### Melanization triggered during the establishment of the latent infection protect the host from supernumerary acute infections

As the protection conferred by ingested PAO1 against a secondary PAO1 acute infection is higher when the challenge is performed at day six, we investigated whether this protection would be active when melanization was impaired in *ΔPPO1-ΔPPO2* mutants. Whereas there was a decreased virulence of injected naïve PAO1 in latently-infected wild-type flies, this was no longer observed in *ΔPPO1-ΔPPO2* mutants (S6A and S6B Fig). This result is however difficult to interpret given the transitioning virulence phenotype of PAO1 colonizing bacteria in *ΔPPO1-ΔPPO2* mutants (S4 Fig).

To bypass this hindrance, we first assessed whether colonizing PAO1 protect to some degree against supernumerary infections with other pathogens. We thus tested using an injection paradigm a mildly pathogenic strain of *S. marcescens*, another Gram-negative pathogen, *Enterococcus faecalis*, *Listeria monocytogenes*, both Gram-positive bacterial pathogens, the latter being also able to induce the IMD pathway due to the presence of diamino-pimelic peptidoglycan in its cell wall, *Metarhizium robertsii*, an entomopathogenic pathogen, and *Candida albicans*, an opportunistic pathogenic yeast. The prior establishment of a PAO1 latent infection conferred a significant degree of protection against all of these infections (Figs 5A–5C and S6C and S6D).

The humoral and the cellular immune responses do not play a predominant role in the protection afforded by PAO1 bacteria that have colonized the *Drosophila* internal tissues (S5D Fig), leaving melanization as a candidate for a host defense mediating protection against secondary infections. As melanization mutant flies are already succumbing fast to the ingestion of PAO1 (Figs 3C and S3D), making it difficult to ascertain precisely the contribution of the secondary infection to the lethality of these PAO1-latently infected flies, we decided to attenuate the virulence of ingested PAO1 by adding levofloxacin instead of gentamicin to a sucrose

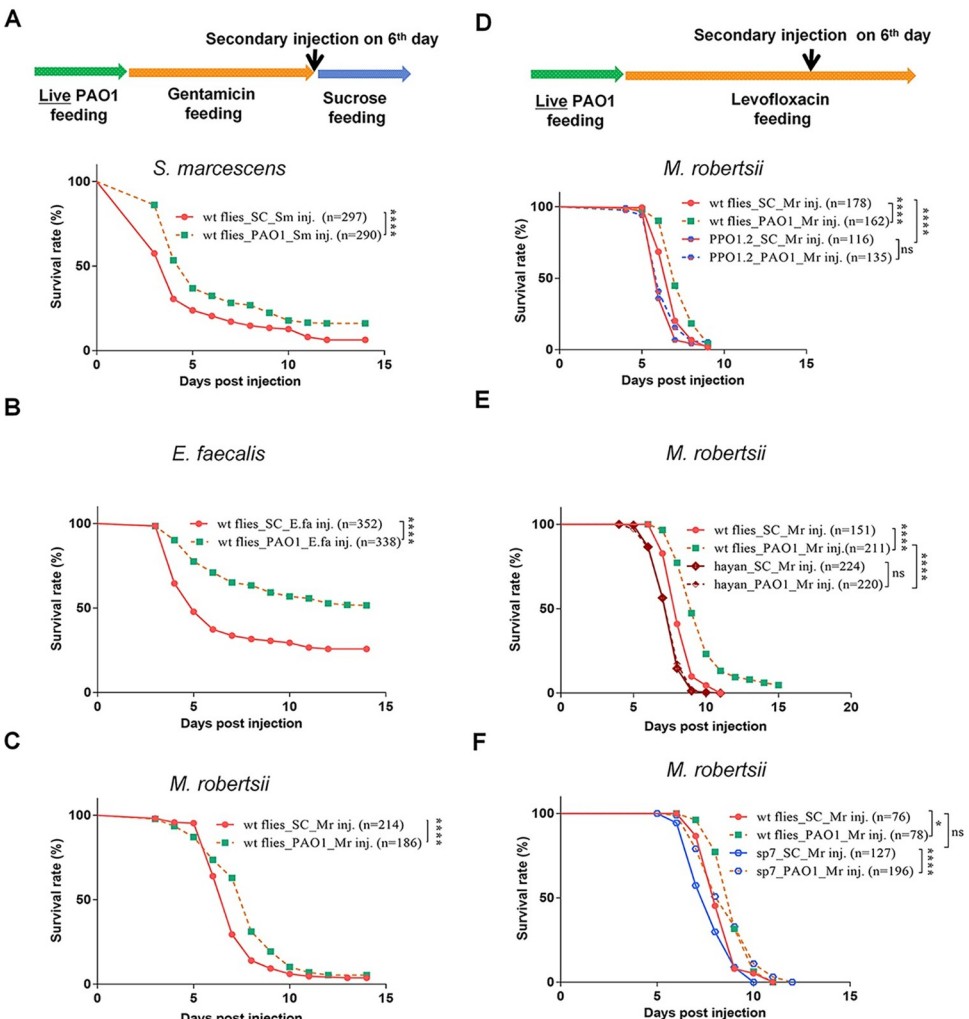

**Fig 5. The silent colonization by PAO1 in the tissues confers to the latently-infected flies an enhanced protection against secondary bacterial or fungal infections that is likely mediated by melanization. (A-C)** Survival curves of secondary infection by the Gram-negative bacterium *Serratia marcescens* (A), the Gram-positive *Enterococcus faecalis* (B), and the fungal entomopathogen *Metarhizium robertsii* in PAO1-latently infected flies as compared to the same infection in naïve flies fed with sucrose solution (C). **(D-F)** Impact of mutations impairing different enzymes involved in melanization on the enhanced defense conferred by the prior ingestion of PAO1: *ΔPPO1-ΔPPO2* (D), *Hayan* (E), and *Sp7* (F). Because these mutants are susceptible to the ingestion of PAO1, the virulence in immunodeficient flies of PAO1 bacteria colonizing the tissues was mitigated by the oral administration of 100μg/mL Levofloxacin for four days, following the ingestion of PAO1 for two days. The *M. robertsii* fungus is insensitive to the action of the antibiotics. All experiments here were performed three times independently and data were pooled together. Statistical analysis was done using the Logrank test (Mantel-Cox test) in (A-F).

solution after two days of PAO1 ingestion. This levofloxacin treatment strategy worked since melanization-defective flies that had ingested PAO1 for two days prior to levofloxacin ingestion did not succumb for at least six days (Fig 5D–5F), in contrast to the situation when these flies are feeding on gentamicin (Fig 3C; this is especially visible for the *ΔPPO1-ΔPPO2* double mutant). Of note, the virulence of PAO1 ingested by melanization-deficient flies treated subsequently with levofloxacin could not be reactivated by PBS injection (S6F Fig), in contrast to the situation with the gentamicin feeding model (S6E Fig). When we checked whether bacteria were still present in wild-type flies treated for five days with *per os* levofloxacin following the

ingestion of PAO1 for two days, we found that bacteria had actually been cleared in half of the tested flies, the other half displaying a reduced titer (S6G Fig), although another set of data (S2D" Fig) showed that the colonizing bacteria were tolerant to levofloxacin for at least two days. Similarly, bacteria were cleared in some 60% of the *ΔPPO1-ΔPPO2* double mutant flies, likely because they fail to colonize the tissues and do not become dormant and are thus susceptible to the levofloxacin treatment (S6H Fig). Thus, the state of dormancy may be transient when only levofloxacin is fed to flies instead of gentamicin.

We next needed to use a pathogen that is not sensitive to this antibiotic, namely a fungus. We chose *M. robertsii*. We then challenged PAO1 latently-infected *ΔPPO1- ΔPPO2*, *Hayan*, or *Sp7* PAO1 mutant flies by the injection of *M. robertsii* conidia on day six, at the end of the levofloxacin treatment. As reported above, the wild-type control flies that had only fed on sucrose solution (laced with levofloxacin after day two) died faster from a *M. robertsii* challenge than flies that had been fed PAO1 and then levofloxacin prior to infection (Fig 5D–5F). As has been found earlier (Jianwen Yang, personal communication), *ΔPPO1-ΔPPO2*, *Hayan*, but not *Sp7* mutant flies fed solely on the antibiotic solution succumbed faster than wild-type flies to the injection of *M. robertsii* conidia, suggesting that *M. robertsii* is not susceptible to the Sp7 arm of melanization (Fig 5D–5F). Strikingly, the protection conferred against *M. robertsii* by the prior ingestion of PAO1 was abolished in the *ΔPPO1-ΔPPO2* and *Hayan* mutants (Fig 5D and 5E), but not in the *Sp7* mutants (Fig 5F), even though the latter are much more sensitive than *Hayan* to the PAO1 latent infection protocol (Fig 3C). It follows that flies are protected to some degree from *M. robertsii* secondary infection in *Sp7* mutants through the activation by PAO1 of the PPO/Hayan axis of melanization, which is presumably not affected in *Sp7* mutants. Even though fed-bacteria may only transiently colonize host tissues when flies are subsequently fed levofloxacin, PAO1 ingestion triggers both the *Hayan* and *Sp7*-dependent branches of melanization and thus provides a broad level of protection against opportunistic secondary infections, according to the specific sensitivities of the supernumerary pathogens to the different facets of the melanization host defense.

## Discussion

In this work, we have established a latent infection model in which ingested *P. aeruginosa* escapes from the digestive tract and colonizes the tissues present within the internal cavity of its *Drosophila* host. It does not cause obvious symptoms except for a sporadic reactivation of its virulence program in a few flies. Importantly, this infection model relies on a passage of bacteria through the gut, a mostly hostile environment for microorganisms characterized by local chemical, biochemical, and immune defenses. In our initial studies, only a few bacteria had been shown to be able to escape into the hemolymph from the adult gut compartment, without significantly damaging it. Here, we document a more extended passage and the formation of a so far unidentified population of bacteria colonizing the host tissues and that are likely adherent since they can no longer be detected in the hemolymph after a few days. Physiologically, the bacteria appear to be dormant, a phenomenon induced to a large extent by the melanization arm of host defense.

### The latent infection established by the oral route differs from that resulting from systemic infections

To our knowledge, only *S. marcescens* and *P. aeruginosa* are able to effectively cross the intestinal barriers (including the peritrophic matrix) and then remarkably display a much-reduced virulence in intestinal infection models [17,20,23]. Persistent infections have been achieved in acute septic injury or injection models with microorganisms of low virulence such as *Candida*

*glabrata* or *Aspergillus fumigatus* [27, 28] or of intermediate virulence, including *Enterococcus faecalis*, *Providencia burhodogranariea*, *Providencia rettgeri*, *Enterococcus lactis* or low virulence *Serratia marcescens* strains [29–33]. However, in these cases, except for the fungal opportunistic pathogens cited above, a significant fraction of infected flies succumb to the challenge and a persistent infection is established solely in flies that have managed to control the infection and therefore survived it. Furthermore, the bacteria that persist may also have developed a strategy to escape the action of the immune system, as proposed by a recent theoretical modeling approach to account for chronic/latent infection [34]. Note that the bacteria strains listed above have been initially isolated from wild infected flies. It is a possibility that they had originally been infected through the oral route, which might be a more frequent way of contamination than injuries. Indeed, *E. faecalis* is a member of the *Drosophila* microbiota [35]. Future studies will tell whether persistent bacteria in these models share morphological and physiological features with *P. aeruginosa*, and possibly *S. marcescens* [23], that have escaped from the digestive tract.

Of note, it has been recently reported that injury alone provides a significant degree of protection against a secondary *E. faecalis* injury, which is mediated to a large extent by hemocytes [26]. Here, there is no injury of the external cuticle in our latent infection model and melanization, not the cellular response, appear to mediate the protection against the secondary challenge with this Gram-positive bacterial pathogen: it depends on the ingestion of living PAO1 bacteria. Thus, priming against secondary infections may be achieved via distinct mechanisms.

## The tissue-colonizing bacteria present specific morphological and physiological properties

The bacteria that colonize the hemocoel present phenotypes that are strikingly distinct from those in an acute infection model or *in vitro* growth such as bacterial morphology, exposure of LPS O-antigen, tolerance to antibiotic treatment, adherence to tissues or motility. Importantly, they do not appear to proliferate. We however cannot formally exclude a slow growth or moderate proliferation of the colonizing bacteria that would be counteracted by immune defenses such as the cellular arm of the innate immune response, resulting in null net proliferation. We can nevertheless exclude a strong proliferation of tissue-associated bacteria as the *Drosophila* immune system is tuned to sensing peptidoglycan fragments released during the cell wall remodeling that occurs during bacterial growth or division since the IMD pathway gets activated even when there is little apparent proliferation (Figs 1G and S1D) [36,37]. Whereas we see a modest induction of the IMD pathway during the establishment of the latent infection likely due to the presence of bacteria in the gut lumen (S5F Fig), we were not able to detect a specific systemic signal using the p*Dipt-GFP* transgenic reporter, except in rare flies in which virulence is spontaneously reactivated.

The properties of these bacteria suggest that they may be dormant, possibly with reduced metabolic activity and growth since they are tolerant to tobramycin, which targets bacterial protein synthesis, and to levofloxacin which inhibits DNA synthesis. Thus, they share many of the characteristics of bacterial persister cells, a minute fraction of a bacterial population able to tolerate antibiotics treatments [2]. A significant difference however is that a large majority of the bacteria that have managed to reach the hemocoel appear to be associated with tissues and dormant with fewer bacteria being planktonic in the hemolymph and likely eliminated through hemocyte-mediated phagocytosis [17,20], in contrast to the few persister cells not killed by antibiotic treatments as observed in *in vitro* and *in vivo* models [2].

The loss of the O5 antigen may alter the properties of the bacterium with respect to host defenses. We have previously reported that *S. marcescens* bacteria deficient in the export of their O-antigen became susceptible to the action of the IMD pathway, likely by making them

sensitive to the action of AMPs [38]. It has been reported that Eater acts as a phagocytosis receptor for Gram-positive but not for Gram-negative bacteria [39], in contrast to our previous report that Gram-negative bacteria such as *S. marcescens* or *P. aeruginosa* are controlled by Eater in intestinal infection models [20,40].This discrepancy might be accounted for by the lack of O-antigen in bacteria that have crossed the intestinal barrier and that might make peptidoglycan more accessible to the Eater receptor, possibly in conjunction with the action of AMPs, in keeping with the proposal that Eater binds to this cell wall component [41].

How the switch in physiological properties of the bacteria occurs remains to be established. Based on the virulence properties of the colonizing bacteria in immune-deficient flies, it appears that the host defense is able to influence the bacterial physiology of bacteria that have escaped from the digestive tract.

## Dormancy is actively induced by the melanization arm of host defense

The study of immune-deficient flies revealed that the induction of a dormant state results from an active action of the host. In the case of persister cells in mammalian infection models, the description of an action of innate immunity in triggering the formation of persister pathogens remains mostly limited to intracellular pathogens undergoing adverse conditions in the phagosomes of macrophages [2]. In the case of *Drosophila*, a humoral immune response, melanization, clearly plays a paramount role. An important observation is that this dormant state correlates to a distinct morphology of the bacteria, both at the single cell and colony levels. It is likely that the less slender form retrieved in the hemolymph during the initial phase of the latent infection model results from a passage through the gut and/or escape into the hemocoel. However, we note that the bacteria appear to keep their more rounded morphology upon reactivation by the injection of PAO1 (S2A Fig), at a time where both the ingested and injected bacteria were actively proliferating (Figs 4C and S5C).

An involvement of melanization and PPOs in inducing or selecting dormant bacteria has been clearly established in this work. The finding that bacteria proliferate as early as day one in the carcass of *ΔPPO1-ΔPPO2* mutants (Fig 3E) suggests that melanization is required to establish the silencing of the virulence program of PAO1. Our current data suggest that ingested PAO1 in the carcass of *ΔPPO1-ΔPPO2* mutants may be only partly dormant, with features of both dormant and virulent states. Thus, melanization is not the sole factor required for establishing dormancy and some input from the cellular or humoral responses is likely. Alternatively, the passage through the gut barrier may also contribute to the establishment of dormancy. Of note, the finding of a reactivation of the virulence program by a simple injury of the cuticle at a time when melanization has already been activated (Fig 4A) suggests that this silencing can be relieved, but only to some extent since "reactivated" bacteria are not as virulent as those in the injection model of naive flies (Figs 4A and 2H').

It is not clear exactly where the melanization pathway gets activated as we did not observe any melanized tissues: such a melanized hindgut had been observed in p38 *Drosophila* mutants [42]. Some *PPO* genes such as *PPO1* and *PPO3* are expressed in the naive gut, with a strong expression of *PPO1* in enteroendocrine cells of the gut proximal regions R1 and R2 and a limited expression of *PPO3* in enteroblasts/enterocytes as well as in visceral muscles (http://flygutseq.buchonlab.com) [43]. Of note, the expression of *PPO* genes was not detected in a midgut single cell RNAseq project [44]. It is also not known how these PPOs would be secreted by gut epithelial cells and whether they would be released apically or basally. Nevertheless, the strongest phenotype is observed with *PPO1-PPO2* mutants and not with single *PPO* mutants. Since *PPO2* is hardly expressed in the gut, it follows that melanization is unlikely to work in the gut lumen.

At present, we hypothesize that bacteria that escape the gut lumen by traversing the intestinal epithelium, either through an intracellular or a paracellular route, may activate the melanization cascade when reaching the hemocoel. How melanization is triggered by Gram-negative bacteria is poorly understood. It has previously been reported that the overexpression of the PGRP-LE peptidoglycan binding-protein triggers the melanization cascade in larvae [45]. However, whether PGRP-LE is required for melanization has not been clearly established, in as much it is now thought to act mostly as an intracellular sensor of short peptidoglycan fragments [37], whereas melanization is an extracellular event. As *P. aeruginosa* can also trigger the Toll pathway in addition to the IMD pathway [16], the proteolytic cascades that activate the Toll pathway likely also trigger the proteases that ultimately process PPOs into active POs [8,46]. Alternatively, the change that affect the cell wall such as the loss of the O5 antigen may also alter the way PAO1 is sensed by the immune system, in keeping with its active avoidance of triggering melanization during acute systemic infections [17].

## Differential actions of melanization effectors according to the targeted pathogen

In terms of effectors, our experiments document a major role for PO1/PO2 and the Sp7 protease but not Hayan. Previous work with a *Staphylococcus aureus* low inoculum injection model has revealed a requirement of PPO1 and Sp7 for an uncharacterized killing activity, in addition to the role of melanization in a classical blackening reaction occurring at the wound site [8]. Recent *in vitro* work has documented Sp7 as being the major protease that cleaves PPOs [11]. In contrast to the situation with PAO1, the proliferation and dissemination of *Aspergillus fumigatus* depends solely on *Hayan*, and not on *Sp7* [28]. Further studies will be required to test the requirements for alternative proteases that can cleave PPO2 or both PPOs, respectively Ser7 and MP1.The finding that *Hayan* is required for cross-protection against a secondary *M. robertsii* infection indicates that both the SP7 and Hayan-dependent killing activities are induced upon the ingestion of PAO1 and that pathogens are susceptible to distinct combinations of PO1/PO2 and Sp7/Hayan. How these combinations of POs and proteases specifically kill distinct pathogens remains to be established. It has been proposed that ROS may be generated during the activation of melanization, but this potential mechanism does not at this stage explain the specificity of the killing reactions.

## Priming against secondary infections in chronically-infected flies

We show here that PAO1 latent infection provides protection against a secondary infectious challenge by Gram-negative bacteria (PAO1 *P. aeruginosa*, *S. marcescens*), by Gram-positive bacteria (*E. faecalis*, *L. monocytogenes*) or by fungal infections (*C. albicans*, *M. robertsii*). Whereas the protection against *M. robertsii* is modest, likely due to the high pathogenicity of this fungus, it depends on the melanization enzymes that are involved in host defense against this entomopathogenic fungus, that is, through a resistance mechanism.

A number of recent studies have examined whether chronic infections established after the injection of pathogens or a pre-challenge with killed pathogens can protect surviving flies from supernumerary challenges and reported variable outcomes. For instance, the injection of heat-killed *Pseudomonas entomophila* or *Lactococcus lactis* failed to provide any protection against secondary infections [33]. In contrast, primary infections with *E. faecalis*, *P. rettgeri*, a non-pathogenic strain of *S. marcescens* or of *P. aeruginosa* allowed a better survival to a secondary pathogenic challenge with related or unrelated pathogens. Depending on the system, it involved the cellular immune response and possibly the *imd* pathway, that is resistance

mechanisms, but also likely resilience mechanisms [30–32,47]. In none of these studies was melanization studied.

Finally, it is important to note that infection by the intracellular and highly contagious symbiont *Wolbachia* species protects the carrier from viral infections. The underlying mechanism has not been definitely deciphered yet, although metabolic effects are likely at play [48–51].

### Relevance of the model of latent infection to human health?

*P. aeruginosa* is a major opportunistic pathogen relevant to public health. While it is a main contributor to the morbidity of cystic fibrosis or burn wound patients, it is also present in a sizable fraction of the population (4 to 20%) as a member of the microbiota [52,53]. Its prevalence is however increased in hospital settings, especially intensive care units and also in infants born in hospitals [52,54,55]. Actually, lung infections can originate from *P. aeruginosa* bacteria translocating from the gut [56,57]. This pathogen is also often found in the digestive tract of cancer patients [54] and may cause systemic infections in neutropenic patients with hematologic malignancies [58]. Often, it is thought that especially virulent strains contribute to the demise of immune-compromised patients. Melanization is a host defense found essentially in protostomes. As discussed above, its effector mechanism(s) remain poorly understood at present and it cannot be excluded that vertebrates harbor a host defense that would rely on similar killing activities, *e.g.*, ROS-mediated killing. It will be thus worth investigating whether *P. aeruginosa* is able to translocate from the gut lumen to the internal milieu in immune-competent vertebrate organisms where it would remain dormant as described here for *Drosophila*. We note that *P. aeruginosa* becomes problematic during surgery [59–61] and one possibility is that silent colonizing bacteria might be "reactivated" by the trauma inherently associated with surgery [61].

## Material & methods

### Experimental model and subject details

**Flies.** The wild type fly strain used throughout the experiments is *w[A5001]*, which was generated from re-isogenized *w$^{1118}$* flies [62]. Most of the fly mutant stocks used in this study have been generated in a *w$^{1118}$* flies background [64]. *PPO1$^\Delta$*, *PPO2$^\Delta$* and *PPO1$^\Delta$-PPO2$^\Delta$* flies [65], *Sp7* and *Hayan* mutant fly stocks [8] were kind gifts from Bruno Lemaitre. *key* and *eater$^\Delta$* flies and the *Diptericin* reporter flies expressing the green fluorescence protein driven by promoter of *Diptericin* are kept in our laboratory. *w[A5001]* is the isogenic control flies for the *key* and *eater$^\Delta$* mutant flies.

**Bacterial strains and culture conditions.** The wild type *P. aeruginosa* strain used in this study is the laboratory reference strain PAO1 and the *wt* PAO1 labeled with GFP or RFP that were constructed to observe *P. aeruginosa* location *in vivo* were kind gifts from Dr. Xiaoxue Wang (Guangzhou, China). The *gfp* or *rfp* coding sequence under basal *accC1* (gentamicin resistance gene) promoter sequences was inserted 15 bp downstream of the *glmS* (15bp) housekeeping gene. Unless mentioned otherwise, all the experiments were done using PAO1 or mutants in the PAO1 background. The other *wt P. aeruginosa* strain used here is another reference strain PA14. All the bacterial strains used in this study are noted in Table 1. Gram-negative bacteria *Serratia marcescens* and Gram-positive bacteria *Listeria monocytogenes* were kind gifts from Dr. Renjie Jiao (Guangzhou, China). Gram-positive bacteria *Enterococcus faecalis* (ATCC19433) was a kind gift from Dr. Garsin and Dr. Lorenz (Houston, USA). *Metarhizium robertsii* a kind gift from Dr. Chengshu Wang (Shanghai, China). All the bacterial stocks were kept at -80˚C refrigerator; bacteria from the frozen stocks were plated on Luria-Bertani agar plate and cultured at 37˚C incubator overnight before use. A single fresh colony was

**Table 1. Key resources table.**

| REAGENT OR RESOURCE | SOURCE | IDENTIFIER |
|---|---|---|
| **Bacterial and fungi strains and plasmids** | | |
| *Pseudomonas aeruginosa* PAO1 | Kind gifts from Dr. X. Wang | N/A |
| PAO1*glmS::gfp* | | N/A |
| PAO1*glmS::rfp* | | N/A |
| *Pseudomonas aeruginosa* PA14 | This study | N/A |
| *Serratia marcescens* #1.2818 | Kind gifts from Dr. R. Jiao | N/A |
| *Listeria monocytogenes* 10403S | | N/A |
| *Enterococcus faecalis* ATCC 19433 | Kind gift from Dr. Garsin and Dr. Lorenz | N/A |
| *Metarhizium robertsii* ARSEF23 | Kind gift from Dr. C. Wang | N/A |
| Fly stocks | | |
| *w* [A5001] | [62] | N/A |
| *kenny* | [20] | N/A |
| *eater*^Δ | [40] | N/A |
| *PPO1*^Δ | Kind gift from Bruno Lemaitre | N/A |
| *PPO2*^Δ | Kind gift from Bruno Lemaitre | N/A |
| *PPO1*^Δ-*PPO2*^Δ | Kind gift from Bruno Lemaitre | N/A |
| *Sp7*^Δ | Kind gift from Bruno Lemaitre | N/A |
| *Hayan*^Δ | Kind gift from Bruno Lemaitre | N/A |
| pDpt-GFP | generated in our laboratory | N/A |
| Antibodies and Enzymes | | |
| Antibody against PA14, rabbit | [17] | N/A |
| Antibody against *Anopheles gambiae* PPO2, rabbit | [63] | N/A |
| Antibody against O5, mice | Biorbyt | orb234239 |
| Goat anti-mouse IgG; DyLight™-conjugated | Thermo Scientific | 35502 |
| Goat anti-rabbit IgG-HRP | Southern Biotech | 1021–05 |
| Goat anti-rabbit IgG (H+L), DyLight™-conjugated | Thermo Scientific | 355552 |
| Commercial Kits | | |
| HiScript III 1st Strand cDNA Synthesis Kit (+gDNA wiper) | Vazyme | R211-01 |
| ChamQ SYBR qPCR Master Mix (Low ROX Premixed) | Vazyme | Q311-02 |
| oligonucleotides | | |
| Rpl49 Fw | Tsingke | GACGCTTCAAGGGACAGTATCTG |
| Rpl49 Rv | Tsingke | AAACGCGGTTCTGCATGAG |
| Diptericin Fw | Tsingke | GCTGCGCAATCGCTTCTACT |
| Diptericin Rv | Tsingke | TGGTGGAGTGGGCTTCATG |
| Attacin Fw | Tsingke | GGCCCATGCCAATTTATTCA |
| Attacin Rv | Tsingke | AGCAAAGACCTTGGCATCCA |
| Cecropin Fw | Tsingke | ACGCGTTGGTCAGCACACT |
| Cecropin Rv | Tsingke | ACATTGGCGGCTTGTTGAG |
| Metchnikowin Fw | Tsingke | CGTCACCAGGGACCCATTT |
| Metchnikowin Rv | Tsingke | CCGGTCTTGGTTGGTTAGGA |
| Hayan Fw | Tsingke | CGGACTATTCCGGCAGTAGT |
| Hayan Rv | Tsingke | CCCGCAGCAGGATCTTTGAT |
| Sp7 Fw | Tsingke | CGGTTTGTTTGCCTTTGGTA |
| Sp7 Rv | Tsingke | ATCGCTGCTTTATGGTGCTC |
| PPO1 Fw | Tsingke | CATCTCCAGAATGCCCCCTT |
| PPO1 Rv | Tsingke | CCTGATTACGCTCATCCACC |
| PPO2 Fw | Tsingke | TGCGGGAGGAGTCTTTTGTG |

*(Continued)*

**Table 1.** (Continued)

| REAGENT OR RESOURCE | SOURCE | IDENTIFIER |
|---|---|---|
| **Bacterial and fungi strains and plasmids** | | |
| PPO2 Rv | Tsingke | TGCCAGTGGTGAAGGTTGAT |
| Chemicals and Antibiotics | | |
| Trizol | Takara | 9109 |
| Trichloromethane | Guangzhou Chemical Reagent Factory | N/A |
| Isopropanol | Guangzhou Chemical Reagent Factory | N/A |
| Ethanol | Guangzhou Chemical Reagent Factory | N/A |
| Diethyl pyrocarbonate treated water | Sangon Biotech | B501005-0500 |
| Congo Red | Amresco | 0379 |
| Coomassie brilliant blue | Sigma | B1131 |
| Hoechst | Thermo Fisher Scientific | H21492 |
| 4% paraformaldehyde | Biosharp | BL539A |
| Carbenicillin | Macklin | C805408 |
| Gentamicin | Amresco | 0304 |
| Levofloxacin | Rhawn | R010691 |
| Tobramycin | Rhawn | R002607 |
| 30% Polyacrylamide | Beyotime | ST003 |
| TEMED | Beyotime | ST728 |
| Tris-Hcl | Boster | AR1162 |
| 100x Protease inhibitor cocktail | Beyotime | P1005 |
| Software and algorithms | | |
| Prism | Graphpad | N/A |
| CE design | Vazyme | N/A |
| Other | | N/A |
| LB broth | Huankai Microbial | 028320 |
| LB agar | Huankai Microbial | 028330 |
| BHB | BD | 237500 |
| Millipore pad | Merck | AP1002500 |
| sucrose | Guangzhou Chemical Reagent Factory | N/A |
| Agar | Mym biological Technology | MA0451 |
| Agarose | Genesand | AG801 |
| Calcium chloride | Tianjin Damao chemical reagent Factory | N/A |

picked to inoculate the Brain-Heart Infusion broth overnight and bacteria were harvested by centrifugation. The bacteria pellet was resuspended and washed in phosphate buffered saline twice prior to its use.

## Method details

### Acute infection

The harvested *P. aeruginosa* pellets were suspended into PBS to measure their optical density. The optical density of bacteria suspension was adjusted to $OD_{600}$ 1.0 in PBS and then diluted to 1:1000 for injection. 3-7d old female adult flies were picked at random for injection, and then 13.8nL of this prepared bacteria suspension (about 100 bacteria) was injected into thorax of each fly by the Nanoject III (Drummond, 3-000-032, USA). The infected flies were put in the artificial climate chamber with temperature of 18°C and 60% humidity and counted every 12 hours until flies died out.

## Latent infection

Before infection, 3-7d old female adult flies were fed100 mM sucrose solution on Millipore pad for 2 days at 25˚C. The harvested *P. aeruginosa* pellets were suspended into 100mM sucrose solution to measure optical density. The optical density of bacteria suspension was adjusted to $OD_{600}$ 10.0 in sucrose/10%BHB solution for the next infection experiment. Each tube with 20 flies was exposed to 600μL bacterial solution and put at 18˚C for 2 days. Then, the flies infected by ingestion were transferred to a new tube with 600μL of 100mM sucrose solution supplemented with 100μg gentamicin and kept for 4 days in this vial to kill *P. aeruginosa* cells in the gut lumen. Then the flies were transferred to new tubes with 600μL of 100mM sucrose solution only (maintaining these flies on regular food medium did not increase the virulence). The flies were counted at regular intervals until flies died out. As compared to the 18˚C condition, the use of a temperature of 25˚C for this procedure led to a more pronounced rate of death (50% death after two weeks) as compared to flies feeding on control solution (S6I Fig), an observation that suggests that temperature is potentially a parameter that regulates virulence.

## Bacterial titer in hemolymph detection

The empty capillary needle was fixed into the Nanoject III machine (Drummond, 3-000-032, USA), then was pricked on the thorax of each fly to collect hemolymph on basis of a capillary effect. The collected hemolymph from each fly was diluted into 10μL prepared PBS in 1.5mL Eppendorf tube. Samples collected above were diluted by a series of 2-fold dilution and then plated on LB agar. These plates were put in an incubator at 37˚C overnight to count the colony forming units.

## Bacterial titer in carcass

The infected flies were anesthetized by carbon dioxide and rinsed with 75% ethanol and PBS. They were then dissected under microscopy to remove heads, guts, Malpighian tubules and ovaries; the remaining tissues of each fly were put in the Eppendorf tube containing 50ul PBS. The tissues were crushed in the Mixer Mill MM400 (Retsch, Germany) by 30 Hertz for 10sec. The samples prepared above were diluted by a series of 2-fold dilution and then plated on LB agar. These plates were put in a 37˚C incubator overnight to count the colony forming units.

## Bacterial titer in whole flies

Each anesthetized fly was put into 1.5ml Eppendorf tube with 50ul PBS inside, and the flies were then crushed using the Mixer Mill MM400 (Retsch, Germany) by 30 Hertz for 10sec.The samples prepared above were diluted by a series of 2-fold dilution and then plated on LB agar. These plates were incubated at 37˚C overnight to count the colony forming units.

## RNA extraction and reverse transcription

Five anesthetized flies or single anesthetized flies (depending on aims of experiments) was collected into one Eppendorf tube with 200μL Trizol for each sample, which was crushed by adding several 2mm grinding zirconium beads and subsequent shaking in the Mixer Mill MM400 (Retsch, Germany). Then, 800μL Trizol was supplemented to each tube for next RNA extraction. For the ground samples in Trizol, 200μL chloroform was added into each tube and mixed adequately; then, the two phases were separated by centrifugation at 12000g for 15min. The upper layer containing the RNA was transferred into a new Eppendorf tube to which 1000μL isopropanol was added to precipitate the RNA. The RNA was pelleted by centrifugation with

12000g for 10min. Then, the RNA pellet was washed by 75% ethanol once and then dissolved in DEPC water. The RNA concentration was measured using a NanoDrop™ One (Thermo, USA). The RNA samples were reverse transcribed into cDNA by HiScript II 1st Strand cDNA Synthesis Kit (R211-01, Vazyme, China) according to the supplier instructions. In brief, the RNA (no more than 1μg) was mixed with the gDNA wiper and incubated at 42˚C for 2min to remove genomic DNA contamination. The reverse transcription mixture containing random primers, reverse transcription polymerase was added to the purified RNA. The mixture was then incubated at 50˚C for 30min followed by 85˚C for 15sec in a 9600 thermocycler (Biorad, Germany).

## RT-qPCR

Gene expression level in flies was measured by real-time quantitative PCR using fluorescence dye. Primers used for quantification in this study are shown in Table 1. The 5x-diluted RT samples for detection were mixed with ChamQ SYBR qPCR Master Mix (Q311-02, Vazyme, China) according to supplier instructions and then run on CFX 96 (Biorad, USA). The expression level of target genes was normalized by the $2^{-\Delta\Delta Ct}$ method after having checked that the efficiencies of the couple of primers were similar.

## *P. aeruginosa* visualization in vivo by expressing fluorescent proteins

The labeled *wt* PAO1 expressing GFP or RFP were used to challenge flies and then infected flies were dissected and observed under fluorescence microscopy.

## *P. aeruginosa* nucleic acids visualization in vivo by fluorescence

Flies challenged with *wt* PAO1 were dissected and fixed with 4% paraformaldehyde for 30min. The fixed fly tissues were washed in PBS 3 times and then incubated in 1:10000 Hoechst solution diluted with PBS supplemented with 0.1% Triton X-100 for 30min. Then the fly tissues were washed 3 times in PBS prior to mounting on a slide and observation under fluorescence microscopy.

## *P. aeruginosa* visualization in vivo by immunofluorescence

As we do not have an antibody raised against PAO1, we challenged flies with PA14 for immunofluorescence observation. Flies with PA14 infection were dissected and fixed in 4% paraformaldehyde for 30min. The fixed fly tissues were washed in PBS for 3 times and then incubated in 5% BSA suspended in PBS solution for 2 hours. Then, the fly tissue were incubated with the primary antibody against PA14 for 1hour. After three washes in 5% BSA solution, the fly tissue was incubated with the DyLight™-488 labeled fluorescent secondary antibody against rabbit for 1hour. Finally, the fly tissues were washed in 5%BSA solution for 3 times prior to observation under fluorescence microscopy.

## Western blot for detection of PPO cleavage

Hemolymph was collected from infected or control flies by capillarity and then diluted into PBS with a serine protease inhibitor cocktail Beyotime (#P1005). The prepared samples were supplemented with SDS-loading buffer and boiled at 95˚C for 5min. The samples were separated by SDS-PAGE (10% acrylamide/bisacrylamide gel, 29:1; 100V; 3 h). Proteins were transferred onto a PVDF membrane (50 min, 12 V), blocked (5% BSA in 20 mM Tris/HCl pH 7.5, 150 mM NaCl and 0.1% Tween-20 or TBST buffer, 2 h) and incubated (4˚C, overnight) with an antibody raised against *Anopheles gambiae* PPO/PO (1:5000, 5% BSA in TBST buffer, 5

mL). Membranes were washed with TBST (1x, 5 min., RT) and incubated with goat anti-rabbit IgG-HRP (Southern Biotech #1021–05, 10 mL, 1:5,000, 1 h, RT). Membranes were washed with TBST (3x, 20 min., RT), developed using the SuperSignal West FEMTO Max. Sensitivity Substrate (#11859290), and visualized using Amersham Imager 680 (GE Healthcare), equipped with a Peltier cooled Fujifilm Super CCD.

### O-antigen staining

The hemolymph with circulating bacteria was collected by the Nanoject III with empty needles based by capillarity. The collected hemolymph was put in the wells of 8-well slides for 30min to deposit bacteria by sedimentation. The bacteria associated with the tissues were collected just by dissecting infected flies to obtain tissues. The fly tissue with adhering bacteria were fixed with 4% paraformaldehyde for 30min and then washed with PBS solution three times. The samples were firstly incubated with 5% BSA solution for 2 hours, then incubated with the O5 primary antibody (Biorbyt, orb234239) diluted in 5% BSA solution for 1hour. The samples were washed in PBS for 3 times, and then incubated with the DyLight™-488 labeled fluorescent secondary antibody against mice for 1hour. The samples were washed in PBS three times and then observed under fluorescence microscopy.

### *Bacteriostatic assay* in vivo

The acutely infected flies were injected with a series of concentrations of Tobramycin and Levofloxacin at 12 hours post *P. aeruginosa* injection to determine the most suitable dose for experimentation. The flies were injected at 12 hours post *P. aeruginosa* injection with the appropriate dose of antibiotics and then kept in a 18˚C incubator. For latently infected flies, the antibiotics doses for injection used was the same as for acute infections. The flies were sacrificed to measure bacteria titer by crushing whole fly at different time points post antibiotics injection. The crushed products were diluted by 10-fold steps and plated on LB agar to count colony forming units the next day.

## P. aeruginosa *cell morphology* in vivo

### Transmission electron microscopy assay

The hemolymph with circulating bacteria was collected by the Nanoject III with empty capillaries. The samples were then processed by the Guangzhou KingMed Diagnostics Company.

### Fluorescence observation

The *wt* PAO1 expressing GFP and RFP was injected and fed to adult female flies separately. Then the infected flies were dissected and observed under fluorescence microscopy.

**P. aeruginosa *colonies morphology*.**　The acutely and latently infected flies were sacrificed respectively on the 2nd and 6th day post infection to be crushed. The crushed product was diluted by a series of 10-fold steps and plated on Congo Red LB agar using cultured *P. aeruginosa in vitro* as control. The Congo Red LB plates were made by adding 40 μg/mL Congo-red, 20 μg/mL Coomassie brilliant blue into LB agar medium [66]. Plates were incubated at 37˚C or 25˚C for 1 day.

**P. aeruginosa *motility assay*.**　The acutely and latently infected *P. aeruginosa* were isolated as for morphology observation and the bacterial titer was measured by plating on LB agar at the same time. The crushed product was dropped on 0.3% agar plates with 1% tryptone and 0.25% NaCl to measure swimming motility [66]. Of note, 80μg/mL Ampicillin was added to

these plates to inhibit microbiota growth. These plates were then put in 37°C or 25°C incubator for one day.

**Secondary infections by septic injury injection.** Flies were fed with live or killed bacteria in different ways and then were injected on 2nd, 6th, 10th day using different categories of pathogens, including *P. aeruginosa*, *Serratia marcescens*, *Listeria monocytogenes*, *Enterococcus faecalis*, *Metarhizium robertsii*, *Candida albicans*. Dead flies were counted each day and survival curves were plotted using Prism 5.0.

## Statistical analysis

Statistical analysis was performed for all the data using the software GraphPad Prism 6.0 based on the types of data distribution. Briefly, the survival curves were compared by Logrank (Mantel-Cox test), the multiple comparisons were performed using Kruskal-Wallis test coupled to Dunn's multiple comparison post-hoc analysis, and unpaired t-test with Welch's correction was used to compare between two groups. The significance level was indicated by the number of stars. *, $p < 0.05$; **, $p < 0.01$; ***, $p < 0.001$; ****, $p < 0.0001$. ns: not significant.

## Supporting information

**S1 Fig. Ingested *P. aeruginosa* exhibits impaired virulence in *Drosophila* (complementary to Fig 1). (A)** Survival curve of flies feeding continuously on *P. aeruginosa* PAO1. This experiment was done three times independently, all data were pooled together. **(B)** clearance of GFP-labeled PAO1 following the ingestion of gentamicin or sucrose solution. The focus is on visceral muscles (right panels). A few bacteria are visible in the crop at high magnification. **(C)** Bacterial load in digestive tract tissues of latently-infected flies at late stage. Sixteen flies were dissected to pick crops and guts respectively. Bacteria secondarily colonizing the outer part of the digestive tract after having escaped from it likely contribute to the measured bacterial titer. This experiment was only done once. Statistics analysis was done using the t-test. **(D)** IMD pathway activation measured by RT-qPCR by monitoring the inducibility of *imd*-regulated AMP genes (*att*; *Attacins*; *cec*: *Cecropins*; *dpt*: *Diptericin*; *mtk*: *Metchnikowin*). The level of induction induced by a systemic challenge with *Escherichia coli* six hours after injection serves as a reference for a full-blown systemic immune response. This experiment was only done once. **(E)** Visualization by fluorescence microscopy of IMD pathway activation in septic injury systemic infection and latent infection using *Diptericin*-GFP transgenic reporter flies, the fluorescence was observed under fluorescence microscope. This experiment was performed three times, and a representative one is presented. **(F)** Bacterial titer in hemolymph of latently infected flies without gentamicin treatment. This experiment was only done once. The bar for each column indicates the median. **(G)** Bacterial titer in tissue of latently infected flies with or without gentamicin treatment. The bar for each column indicates the median. The t-test was used to assess statistical significance for panels D and G.
(TIF)

**S2 Fig. Features of sessile PAO1 in carcass (complementary to Fig 2). (A)** Morphology of *P. aeruginosa* in distinct infection routes. Flies were fed with $OD_{600}$ 10 RFP-labeled PAO1 for 2 days and then were injected with GFP labeled-PAO1; flies were then dissected two days post injection to observe bacterial morphology under fluorescence microscopy in the same fly. **(B)** The luminal content of the gut of flies that have ingested PAO1-GFP bacteria was examined. Killed bacteria stained by propidium iodide appear to be red whereas live bacteria are green. **(C-C')** *P. aeruginosa* dose-response inhibition by the injection (4.6 nL) of different doses of Tobramycin (TOB) (C) and of Levofloxacin (LEV) (C'). **(D-D")** TOB or LEV *per os* treatment

in *P. aeruginosa* acute (D and D') and latent infection (D"). Survival curve of the flies with PAO1 acute injury infection feeding afterwards on TOB and LEV(D). Bacterial titer of PAO1 latently-infected flies feeding on TOB and LEV (D'). Bacterial titer of flies with PAO1 acute injury infection feeding afterwards on TOB and LEV (D"). The bar for each column indicates the median (D' and D"). **(E-E')** Pathogenicity potential of sessile PAO1 in naive flies. *P. aeruginosa* was isolated from carcass of latently-infected flies at different time points and then injected into naive flies. This experiment was done three times and a representative one is presented here. Statistical analysis was done using Logrank (Mantel-Cox test) in (C-D, E-E') and Kruskal-Wallis with Dunn's post-hoc test in (D'-D").
(TIF)

**S3 Fig. Role of different arms of innate immunity against ingested *P. aeruginosa* (complementary to Fig 3). (A)** Survival curves of PAO1 latently-infected flies in which the phagocytic abilities of hemocytes have been saturated by the injection of latex-beads that can be phagocytosed but not digested at the indicated time; PBS injection is a control for the effect of injury. LxB indicates latex-beads. Note that the survival curve starts on the day of LxB injection. **(B)** Bacterial load of PAO1 latently-infected flies with impaired phagocytosis. **(C)** Survival curves of *ΔPPO1-ΔPPO2* immuno-deficient flies after PAO1 acute injury infection. **(D)** Survival curves of *ΔPPO1* or *ΔPPO2* immuno-deficient flies after PAO1 latent infection establishing the redundancy of *PPO1* and *PPO2*. **(E)** Measurement of IMD pathway activation in *PPO* mutant flies using *Diptericin* steady-state levels measured by RT-qPCR as a read-out of its activation. **(F)** Quantification of the intensity of cleavage of PPO into PO in three independent Western blots, one of which is displayed in Fig 3F. **(G-H)** Expression level of melanization genes measured by RTqPCR after the injection of PAO1 (F) or after PAO1 latent infection (G). The experiments were performed three times and the data were pooled together (A-H). Statistics analysis was done by t-test in (B, E-H), and for each column the bar indicates the mean (B, E, F, H).
(TIF)

**S4 Fig. Features of sessile PAO1 in PPO-deficient flies (complementary to Fig 3). (A-B)** Sensitivity of bacteria *in vivo* to 4.6 nL-injected tobramycin (TOB: 16mg/mL) or levofloxacin (8mg/mL) measured by monitoring the survival (A) and assessing the bacterial load of single whole flies of *ΔPPO1-ΔPPO2* immuno-deficient flies (B). Flies were treated with antibiotics on the 3rd day post infection. **(C)** Colony morphology of PAO1 retrieved from *ΔPPO1-ΔPPO2* immuno-deficient flies on 4th day. **(D)** O5 staining of PAO1 in *ΔPPO1-ΔPPO2* immuno-deficient flies. The experiments were performed three times and the data were pooled together (A, B), statistics analysis was done using the Logrank test (Mantel-Cox test) in (A) and the Kruskal-Wallis test with Dunn's post-hoc test in (B); the bar for each column indicates the median (B). The experiments were performed three times and a representative one is presented here for (C, D).
(TIF)

**S5 Fig. Injury or a secondary infection of flies activate the virulence of ingested *P. aeruginosa* that have escaped from the digestive tract (complementary to Fig 4). (A-B)** Injury of flies at Day 6 or Day 10 activates only partially the virulence of ingested *P. aeruginosa*. **(C)** Growth of ingested GFP-labeled PAO1 and injected RFP-labeled PAO1 six days after the beginning of the latent infection protocol. **(D)** The protection afforded by a latent PAO1 infection against a secondary injection of PAO1 at day 2 of the latent infection protocol is still effective in *key-* or *eater*-deficient flies. SC: control flies fed on an uncontaminated sucrose solution **(E)** The protection afforded by the ingestion of UV-killed bacteria against a secondary PAO1

challenge at day 2 after ingestion requires *key* but not *eater*. **(F)** Whole-flies *Diptericin* steady-state expression levels as measured by RTqPCR expression level in flies is induced 48 hours after the ingestion of live or killed PAO1 *P. aeruginosa.* **(G-G')** Protection afforded against a secondary PAO1 injection by the ingestion of either UV-killed (G) or PFA-killed PAO1. All the experiments presented here have been performed three times independently and data were pooled together and analyzed by Logrank (Mantel-Cox test) in (A-B, D-E, G-G'), using the Kruskal-Wallis test with Dunn's post-hoc test in (F); the bar for each column indicates the median (C, F).
(TIF)

**S6 Fig. Protection role of *P. aeruginosa* in hemocoel against other pathogens (complementary to [Fig 5]). (A-B)** Absence of a protective role of ingested *P. aeruginosa* in *ΔPPO1-ΔPPO2* immuno-deficient flies after $10^2$ cfu (A) or $10^3$ cfu (B) PAO1 injection. **(C-D)** Protective role of ingested *P. aeruginosa* against *Listeria monocytogenes* (C) or against *Candida albicans* systemic infection (D). **(E-F)** Survival curves of *ΔPPO1-ΔPPO2* immuno-deficient flies with PAO1 latent infection by different treatment: PBS injection with gentamicin feeding in (E) and PBS injection with levofloxacin feeding in (F). **(G)** Bacterial titer of PAO1 ingested for two days by wild-type flies followed by levofloxacin feeding for 5 days. **(H)** *ΔPPO1-ΔPPO2* deficient flies with levofloxacin feeding for 4 days. **(I)** Survival curve of flies with PAO1 latently-infected performed at 25˚C. Flies were fed with 1 $OD_{600}$ PAO1 suspended in 100mM sucrose solution with 10 percent Brain-Heart Infusion broth for 2 days and then PAO1 in gut lumen were killed by feeding sucrose solution containing 100µg/mL gentamicin for another 4 days, leaving alive only the bacteria that had crossed the digestive tract. Then flies were fed sucrose solution until death. Flies were put at 25˚C throughout the infection process. Data have been analyzed using the Logrank test (Mantel-Cox test) in (A-F, I).
(TIF)

## Acknowledgments

We thank Dr. Xiaoxue Wang for the *P. aeruginosa* strains used in this study, Dr. Renjie Jiao for the *S. marcescens* and *Listeria monocytogenes* strain, Drs. Danielle Garsin and Michael Lorentz for the *E. faecalis* strain, Dr. Chengshu Wang for the *M. robertsii* strain, the late Dr. Hans-Michael Muller for the PO antibody. We are grateful to Dr. Bruno Lemaitre for the kind gifts of *Sp7*, *PPO1^Δ^*, *PPO2^Δ^*, *PPO1^Δ^-PPO2^Δ^* fly strains. We thank Dr. Chuping Cai for help with the design of schemes for figures.

## Author Contributions

**Conceptualization:** Jing Chen, Guiying Lin, Dominique Ferrandon.

**Formal analysis:** Jing Chen, Guiying Lin, Dominique Ferrandon.

**Funding acquisition:** Jing Chen, Zi Li, Dominique Ferrandon.

**Investigation:** Jing Chen, Guiying Lin, Kaiyu Ma, Samuel Liégeois.

**Methodology:** Jing Chen, Guiying Lin.

**Project administration:** Jing Chen, Zi Li, Dominique Ferrandon.

**Supervision:** Jing Chen, Zi Li, Samuel Liégeois, Dominique Ferrandon.

**Writing – original draft:** Jing Chen, Dominique Ferrandon.

**Writing – review & editing:** Jing Chen, Guiying Lin, Samuel Liégeois, Dominique Ferrandon.

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
