## [Decision Letter · Decision Letter 0]

10 May 2024

Dear Dr Ferrandon,

We are pleased to inform you that your manuscript 'A specific innate immune response silences the virulence of Pseudomonas aeruginosa in a latent infection model in the Drosophila melanogaster host' has been provisionally accepted for publication in PLOS Pathogens.

Best regards,

David Skurnik, M.D., Ph.D.

Section Editor

PLOS Pathogens

David Skurnik

Section Editor

PLOS Pathogens

Michael Malim

Editor-in-Chief

PLOS Pathogens

orcid.org/0000-0002-7699-2064

Reviewer Comments (if any, and for reference):

Reviewer's Responses to Questions

**Part I - Summary**

Reviewer #1: I have reviewed this manuscript for Review Commons and would like to thank the authors for their efforts to address my comments.

I have no further concerns.

**Part II – Major Issues: Key Experiments Required for Acceptance**

Reviewer #1: (No Response)

**Part III – Minor Issues: Editorial and Data Presentation Modifications**

Reviewer #1: (No Response)

PLOS authors have the option to publish the peer review history of their article (what does this mean?). If published, this will include your full peer review and any attached files.

Reviewer #1: No

---

## [Editor Report · Acceptance letter]

30 May 2024

Dear Dr Ferrandon,

We are delighted to inform you that your manuscript, "A specific innate immune response silences the virulence of Pseudomonas aeruginosa  in a latent infection model in the Drosophila melanogaster host," has been formally accepted for publication in PLOS Pathogens.

Best regards,

Michael Malim

Editor-in-Chief

PLOS Pathogens

orcid.org/0000-0002-7699-2064